# lncRNA transcription induces meiotic recombination through chromatin remodelling in fission yeast

Satoshi Senmatsu[1], Ryuta Asada[1,5], Arisa Oda [2], Charles S. Hoffman [3], Kunihiro Ohta[2,4] & Kouji Hirota [1✉]

Noncoding RNAs (ncRNAs) are involved in various biological processes, including gene expression, development, and disease. Here, we identify a novel consensus sequence of a *cis*-element involved in long ncRNA (lncRNA) transcription and demonstrate that lncRNA transcription from this *cis*-element activates meiotic recombination via chromatin remodeling. In the fission yeast *fbp1* gene, glucose starvation induces a series of promoter-associated lncRNAs, referred to as metabolic-stress-induced lncRNAs (mlonRNAs), which contribute to chromatin remodeling and *fbp1* activation. Translocation of the *cis*-element required for mlonRNA into a well-characterized meiotic recombination hotspot, *ade6-M26*, further stimulates transcription and meiotic recombination via local chromatin remodeling. The consensus sequence of this *cis*-element (*mlon-box*) overlaps with meiotic recombination sites in the fission yeast genome. At one such site, the *SPBC24C6.09c* upstream region, meiotic double-strand break (DSB) formation is induced in an *mlon-box*-dependent manner. Therefore, mlonRNA transcription plays a universal role in chromatin remodeling and the regulation of transcription and recombination.

[1] Department of Chemistry, Graduate School of Science, Tokyo Metropolitan University, Hachioji-shi, Tokyo, Japan. [2] Department of Life Sciences, The University of Tokyo, Meguro-ku, Tokyo, Japan. [3] Biology Department, Boston College, Chestnut Hill, MA, USA. [4] Universal Biology Institute, The University of Tokyo, Bunkyo-ku, Tokyo, Japan. [5] Present address: Department of Viticulture and Enology, University of California, Davis, Davis, CA, USA. ✉email: khirota@tmu.ac.jp

Transcriptome analyses have revealed that numerous non-coding RNAs (ncRNAs) are transcribed in eukaryotic cells[1–3]. Long ncRNAs (lncRNAs) that are more than 200 nucleotides are transcribed from regions with little protein-coding potential such as intergenic and promoter regions, and the relationship between such pervasive transcriptions and various biological processes, such as gene expression, development, disease, and the control of meiotic recombination is of great biological importance[4–13].

In meiosis, homologous recombination between homologous chromosomes is pivotal for meiotic progression and contributes to proper segregation of homologous chromosomes during gametogenesis[14–17]. This process is initiated by the formation of double-strand breaks (DSBs) catalyzed by the Spo11 protein[18]. Distribution of meiotically induced DSBs is not uniform but is clustered at "hotspots" where meiotic recombination is frequently induced[19,20]. The global distribution of DSB has been extensively studied in the highly diverged yeasts *Saccharomyces cerevisiae* (budding yeast) and *Schizosaccharomyces pombe* (fission yeast)[21–24]. High resolution mapping of DSB sites in budding yeast has provided insights into DSB sites and hierarchical context, such as chromosome structures, chromatin, transcription factors, and local sequence composition[25]. Most DSB hotspots are located in promoter regions for mRNA[26] or ncRNA[13], which are characterized by transcription factor binding, histone modifications and low nucleosome density, suggesting a close relationship between meiotic recombination and transcription. In fission yeast, the *ade6-M26* hotspot, which contains a G-to-T substitution in the *ade6* open reading frame (ORF) has been intensively investigated as a model locus of meiotic recombination hotspot[27,28]. The *M26*-mutation is a nonsense mutation creating a cyclic adenosine monophosphate (cAMP)-responsive element (*CRE*)-like heptanucleotide sequence 5′-ATGACGT-3′, to which the transcription factors Atf1-Pcr1 bind and activate meiotic recombination[29–32]. Atf1-Pcr1 binding induces *M26* transcription from the *M26*-mutation site in the *ade6* ORF, followed by histone acetylation and chromatin remodeling around the *M26* site, thereby inducing meiotic recombination[31–33]. Taken together, these observations suggest that local chromatin configuration, histone modifications and transcriptional activity play key roles in the location of meiotic DSB sites in budding yeast and fission yeast[13,32–36]. However, mechanisms underlying the determinants of this DSB distribution have not been fully elucidated.

Previously, we found a stepwise transcription of lncRNAs in the upstream region of fission yeast *fbp1* gene[37]. The *fbp1* gene is activated upon glucose starvation[38], and this activation is mediated by two transcription factors, Atf1 and Rst2[39,40]. These transcription factors bind to critical *cis*-acting binding sequences located upstream of the *fbp1* promoter, (upstream activating sequences 1 [UAS1] and 2 [UAS2])[39] (Fig. 1a). During glucose starvation stress, several species of lncRNAs, referred to as metabolic stress-induced lncRNAs (mlonRNAs), are transcribed from the upstream region of the *fbp1* promoter[37,41]. Stepwise mlonRNA transcription (mlonRNA-a, mlonRNA-b, and mlonRNA-c initiating progressively closer to the *fbp1* transcription start site [TSS]) induces chromatin remodeling at the *fbp1* promoter (Fig. 1a), and subsequently facilitates transcription factor binding to activate *fbp1* expression[37]. These observations indicate that mlonRNA transcription mediates chromatin remodeling and thereby contribute to the robust induction of *fbp1* transcription upon glucose starvation. However, whether or not the role played by this lncRNA transcription is limited to the *fbp1* locus has not been addressed. In this study, we investigated whether mlonRNA transcription plays a universal role in chromatin remodeling in the fission yeast genome, leading to the discovery that this lncRNA transcription induces meiotic recombination through inducing chromatin remodeling.

## Results

### mlonRNA initiation element (*mlon-IE*) works at other loci and conditions in addition to its role in the *fbp1* upstream region.
We previously identified lncRNAs expressed in the fission yeast *fbp1* upstream region in the process of massive transcriptional activation, naming these RNAs mlonRNA-a, mlonRNA-b, and mlonRNA-c (Fig. 1b). These transcripts initiate from far upstream of the *fbp1* promoter and the initiation site progressively shifts closer to the *fbp1* TSS (Fig. 1a, b). The chromatin configuration was analyzed by the distribution of micrococcal nuclease (MNase) sensitive site in *fbp1* upstream region (Fig. 1b). MNase-sensitive sites appear 10 min after glucose starvation (Fig. 1b, arrowheads), coincident with the appearance of mlonRNA-b transcription. Another MNase-sensitive region appears around UAS1-UAS2 at 20 to 30 min after glucose starvation (Fig. 1b, dashed lines), when mlonRNA-c transcription is first observed. Finally, intense MNase-sensitive sites appear around the TATA box at 60 to 180 min after glucose starvation (Fig. 1b, thick lines), when massive *fbp1* mRNA transcription occurs. Our previous study including these observations indicated that mlonRNA transcription induces chromatin remodeling in the transcribed tract in *fbp1* upstream during the transcriptional activation processes[37]. We further identified the *cis*-element (mlonRNA initiation element [*mlon-IE*]) (5′-ATCTTATGTA-3′) required for initiation of mlonRNA-c transcription[42]. To investigate the generality of the role of mlonRNA transcription in chromatin modulation and to further assess the role of this lncRNA transcription in other aspects of chromosomal regulation, we inserted this *cis*-element into the *ade6-M26* meiotic recombination hotspot (Fig. 1c)[27,28]. The *M26*-mutation is a nonsense mutation creating a *CRE*-like Atf1-Pcr1 bind site[29–32]. Since *mlon-IE* is located 200 bp downstream from UAS1 comprising an Atf1-Pcr1 binding sequence in *fbp1*, and Atf1 is required for mlonRNA-c expression[37], the 65 bp *fbp1* upstream sequence containing *mlon-IE* and a flanking sequence was placed 200 bp downstream from the *M26* mutation site by replacing the same length of the *ade6* ORF sequence (*M26::mlon-IE* cells). During meiosis, an additional shorter transcript is induced in *M26::mlon-IE* cells, but it does not appear in the *M26::mlon-IE-mut* control carrying mutated *mlon-IE* (Fig. 1c, d). The TSS of this shorter transcript is +426 from the *ade6* ORF ATG, located 94 bp downstream of the *mlon-IE* core 10 bp (Supplementary Fig. 1a). These findings show that *mlon-IE* works in other loci and conditions, in addition to its role in the *fbp1* locus under glucose starvation stress.

We also translocated the *cis*-element into a control mutant allele of *ade6*, *ade6-M375*, which has a similar G-to-T substitution without creating a *CRE*-like sequence (Fig. 1c). *M375::mlon-IE* cells do not show transcription from *mlon-IE* (Fig. 1d). Similar to what is observed during meiosis, osmotic stress induces a shorter transcript downstream of the *mlon-IE* insertion site in *M26::mlon-IE* cells but not in *M26::mlon-IE-mut* or *M375::mlon-IE* cells (Supplementary Fig. 1b, c). These findings suggest that *mlon-IE* can also work during osmotic stress, and depends on the binding of the Atf1-Pcr1 transcription factors upstream of *mlon-IE*. This result is consistent with what is observed in the mlonRNA-c initiation during *fbp1* activation, which is dependent on *UAS1* comprising Atf1-Pcr1 binding sequence[37].

### *mlon-IE*-induced transcription activates meiotic recombination via inducing local chromatin remodeling.
We next examined whether transcription from *mlon-IE* activates meiotic recombination. Briefly, we measured the recombination rate in the *ade6* locus using tester strain *ade6-469*, which has a nonsense mutation in the 3' region of the *ade6* gene

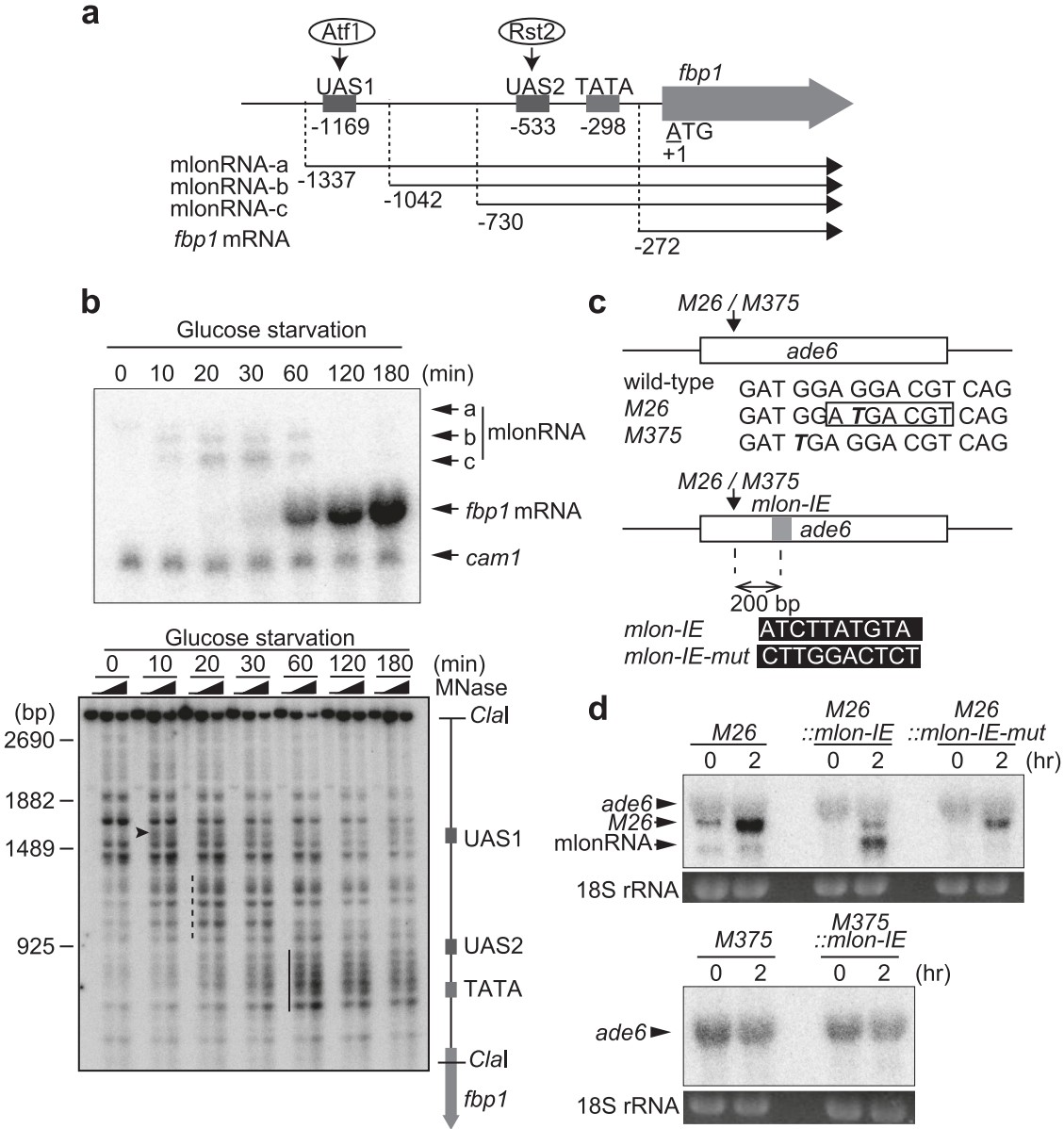

**Fig. 1 *mlon-IE* works at other loci and conditions in addition to its role in the *fbp1* upstream region. a** Schematic representation of the *fbp1* upstream region containing upstream activating sequence 1 and 2 (UAS1 and UAS2), the binding sites for Atf1 and Rst2, respectively. The numbers indicate the transcription start site of the *fbp1* transcripts and the distances of UAS1, UAS2 and the TATA box from the first ATG of *fbp1* open reading frame (ORF). **b** Representative images of northern blot showing stepwise expression of mlonRNAs and *fbp1* mRNA and blot of the chromatin analysis showing stepwise chromatin remodeling during glucose starvation. Wild-type haploid cells were grown to $2.0 \times 10^7$ cells/ml in YER medium, then transferred to YED medium. Cells were harvested at the indicated times. *cam1* transcript was used as an internal control. **c** *M26* and *M375* mutations in *ade6* (bold italic) and inserted *mlon-IE* sequences are shown. **d** *ade6* transcription in indicated cells. Diploid cells were cultured to induce meiosis, as described in the Methods section. 18S rRNA stained by ethidium bromide is shown as the loading control.

(Supplementary Fig. 2a)[32]. *M26::mlon-IE* cells and parental *ade6-M26* cells exhibit a similar number of recombinant Ade+ colonies in which a functional *ade6+* gene is reconstituted by homologous recombination (Fig. 2a). Unfortunately, we cannot directly compare the number of Ade+ recombinants between *ade6-M26* and *M26::mlon-IE* cells for the following reasons. Insertion of *mlon-IE* drastically changes the amino acid sequence of the Ade6 protein and thereby disrupts the protein function (Supplementary Fig. 2b). This insertion makes the distances between *M26::mlon-IE/469* shorter than that between *M26/469* (Supplementary Fig. 2a). Moreover, the *M26::mlon-IE* allele has multiple mutations that would require longer recombination tracts for the reconstitution of a functional *ade6+* gene than needed for the

*ade6-M26* allele. However, *M26::mlon-IE* cells show a significantly higher meiotic recombination rate compared to *M26:: mlon-IE-mut* cells (Fig. 2a). In contrast, *M375::mlon-IE* and *M375::mlon-IE-mut* cells show a similar recombination rate (Fig. 2a). These findings indicate that *mlon-IE* induces mlonRNA transcription even at 200 bp downstream of the *M26* mutation point in the *ade6-M26* gene and that this expression may augment meiotic recombination in comparison to mutated *mlon-IE* inserted cells.

Considering the role *mlon-IE*-induced transcription plays in meiotic recombination, we examined meiotic DSBs, which are introduced by Rec12 (Spo11 ortholog in fission yeast) and initiate meiotic recombination[43]. In *M26* and *M26::mlon-IE-mut* cells, we

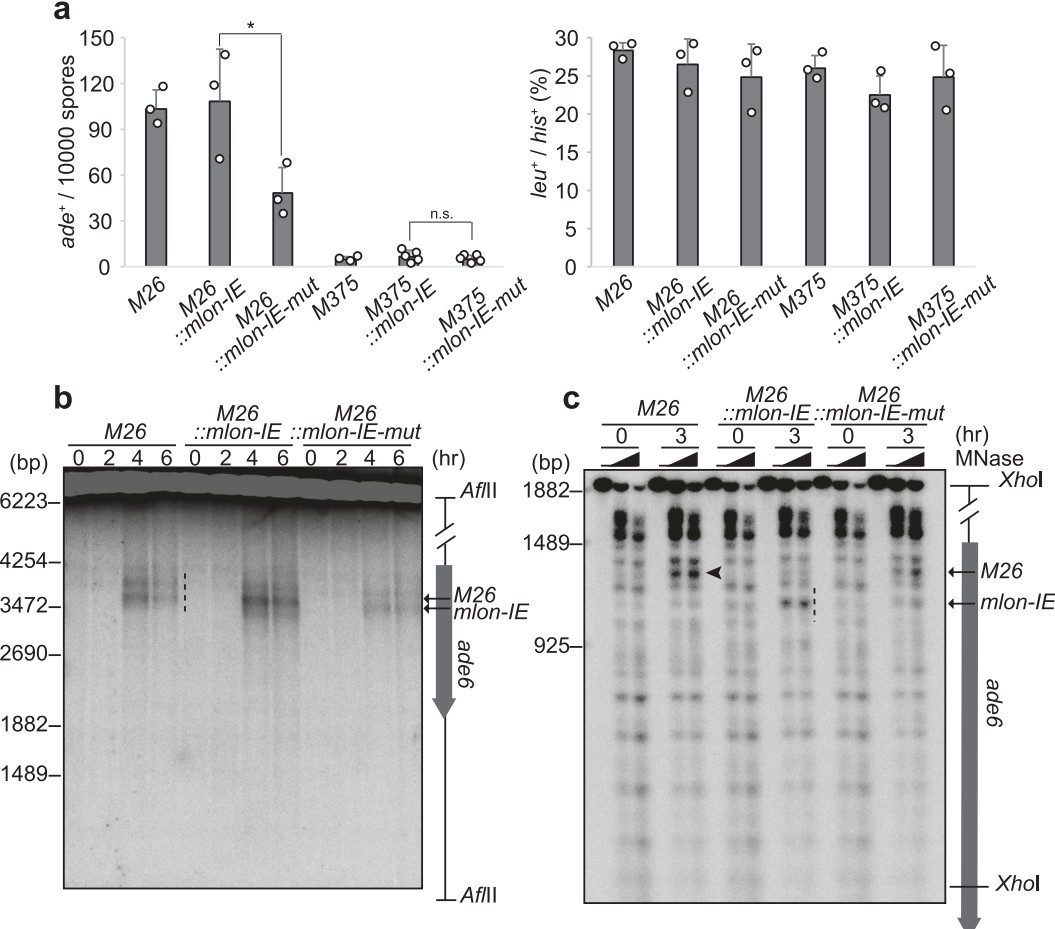

**Fig. 2 Transcription from *mlon-IE* activates meiotic DSB formation and recombination in *ade6-M26*, the meiotic recombination hotspot.**
**a** Recombination rates at the *M26* or *M375* control allele (indicated as *ade*+/10⁴ spores) were examined, as described in the Methods section (left). Recombination between *leu1-32* and *his3-D1* (indicated as *leu*+/*his*+ percentage) was measured as the control (right). Error bars represent standard deviations. $n = 3$ or 5 biologically independent experiments. *P* values were calculated using the unpaired one-sided Student's *t*-test: *$P < 0.05$, n.s. (not significant). **b** Indicated haploid cells possessing the *pat1-114 rad50s* genotype were cultured to induce meiosis, and DNA was prepared as described in the Methods section. Meiotic DSBs were detected by Southern blotting. The dotted line indicates DSBs around *M26* and inserted *mlon-IE*. **c** Meiotic chromatin remodeling in indicated cells. Diploid cells were cultured as in Fig. 1c. The black arrowhead and the dotted line indicate MNase-sensitive sites at the *M26* mutation and the inserted *mlon-IE*, respectively.

detected two DSB sites of comparable intensity (Fig. 2b dotted lines and Supplementary Fig. 3a, b). Remarkably, *M26::mlon-IE* cells show increased intensity of DSBs at the *mlon-IE* site, and the DSB distribution changes with marked bias toward the *mlon-IE* site (Fig. 2b and Supplementary Fig. 3a, b). As genome-wide DSBs assessed by pulse-field gel electrophoresis (PFGE) in these strains are indistinguishable, global DSB formation activity is not affected by the insertion of *mlon-IE* (Supplementary Fig. 4). To determine whether the effect of *mlon-IE*-induced transcription on DSB formation is due to transcription-coupled chromatin remodeling, we analyzed the chromatin configuration by MNase digestion as in Fig. 1b. In all cells carrying the *M26* mutation, the chromatin at *M26* is protected from MNase before meiosis (Fig. 2c arrowhead and Supplementary Fig. 3c), although some sensitive sites are observed nearby. At 3 h after onset of meiosis, MNase-sensitive sites appear at the *M26* site (Fig. 2c arrowhead and Supplementary Fig. 3c)[31,33]. Meanwhile, *M26::mlon-IE* cells show an additional intense MNase-sensitive band at the *mlon-IE*-inserted site (Fig. 2c dotted line and Supplementary Fig. 3d), with a slight reduction in the MNase-sensitive band at the *M26* site. These findings show that the *M26* mutation induces alteration of

the chromatin configuration into the open state in meiosis. More importantly, transcription induced from *mlon-IE* further induces chromatin opening around the transcriptional initiation site. This local alteration of the chromatin geometry around the *mlon-IE* site might influence adjacent chromatin configuration at *M26*. Taken together, *mlon-IE*-mediated transcription induces chromatin remodeling, thereby facilitating meiotic DSB formation and augmenting the meiotic recombination rate.

**_mlon-IE_ works downstream of transcription factor–binding sequences.** Similar to *M26* sequence, other mutations at *ade6* that create transcription factor–binding sequences, including *CCAAT*, *oligo-C*, *4095*, and *4156* motifs, activate meiotic recombination in the *ade6* locus[44]. Next, we tested whether *mlon-IE* works together with these transcription factor–binding sequences instead of the *M26* sequence. In *ade6-4002* (CCAAT-motif) and *ade6-4099* (oligo-C-motif), hotspot activity depends on transcription factors, CCAAT-binding factor CBF (heterotrimer composed of Php2, Php3, and Php5), and $C_2H_2$ zinc finger protein (Rst2), respectively[45]. Regulators of *ade6-4095* and *ade6-4156* hotspot activity are unclear, although the *ade6-4156* sequence has the potential to

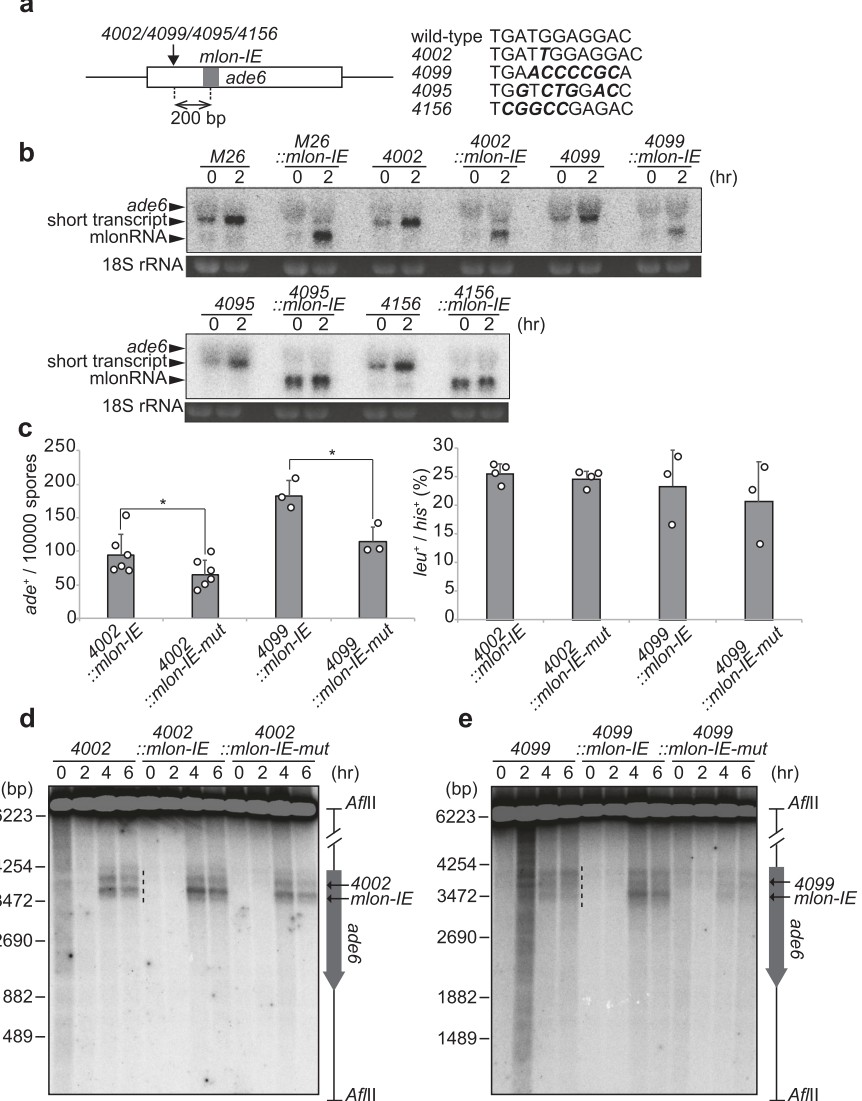

**Fig. 3 mlon-IE works downstream of binding sequences for transcription factors, CCAAT-binding factor and Rst2. a** Schematic representation of *ade6-4002*, *ade6-4099*, *ade6-4095* and *ade6-4156*. *4002* insertion, *4099* mutation, *4095* mutation and *4156* mutation are shown (bold italic). *mlon-IE* was translocated 200 bp downstream from the *4002/4099/4095/4156* mutation. **b** *ade6* transcripts of indicated cells in meiosis. Cells were cultured, as described in Fig. 1c. **c** Recombination rates of indicated cells were assessed, as described in Fig. 2a. Error bars represent standard deviations. *n* = 3 or 6 biologically independent experiments. *P* values were calculated using the unpaired one-sided Student's *t*-test: \**P* < 0.05. **d**, **e** Meiotic DSBs of indicated cells were detected by Southern blotting, as described in Fig. 2b.

bind the Rds1 protein[19]. We inserted *mlon-IE* 200 bp downstream of *ade6-4002*, *4099*, *4095*, and *4156* sequences (Fig. 3a). These sequences induce a shorter transcript from inside the *ade6* ORF, similarly to that seen in *ade6-M26* cells (Fig. 3b). More importantly, *mlon-IE* inserted cells (*4002::mlon-IE*, *4099::mlon-IE*, *4095::mlon-IE*, and *4156::mlon-IE* cells) show an additional shorter transcript (Fig. 3b), indicating that *mlon-IE* also works in the presence of other transcription factor–binding site, in addition to Atf1-Pcr1 binding site. We observed the additional shorter transcript in *4002::mlon-IE* or *4099::mlon-IE* cells only in meiosis (2 h), while that in *4095::mlon-IE* or *4156::mlon-IE* cells was strongly transcribed before the onset of meiosis (0 h; Fig. 3b). This difference might be due to the different binding pattern of transcription factors to the mutated *ade6* loci (*4002*, *4099*, *4095*, and *4156*). i.e., *4002* and *4099* are bound by transcription factors in a meiosis-specific manner, while *4095* and *4156* are constitutively bound by some transcription factors. Similar to *M26::mlon-IE* cells, additional transcriptional initiation from inserted

*mlon-IE* activates recombination in *ade6-4002* and *ade6-4099* (Fig. 3c). In addition, the DSB distribution changes with a bias toward the *mlon-IE*-inserted site in *4002::mlon-IE* and *4099::mlon-IE* cells (Fig. 3d, e and Supplementary Fig. 5a–d). Again, DSB activity on a genome-wide scale is not affected by the insertion of *mlon-IE* in these strains (Supplementary Fig. 4).

**Natural mlonRNA transcription from an intergenic region activates meiotic DSB formation via inducing local chromatin remodeling.** Having established the role of mlonRNA transcription from *mlon-IE* in the activation of meiotic recombination in the *ade6* locus, we next asked if mlonRNA transcription plays a role in the regulation of meiotic recombination through chromatin modulation in the *S. pombe* genome. To this end, we determined the consensus sequence of *mlon-IE* by comprehensively mutating each of the 10 nucleotides existing in *mlon-IE* (Fig. 4a). By excluding mutations that decrease mlonRNA

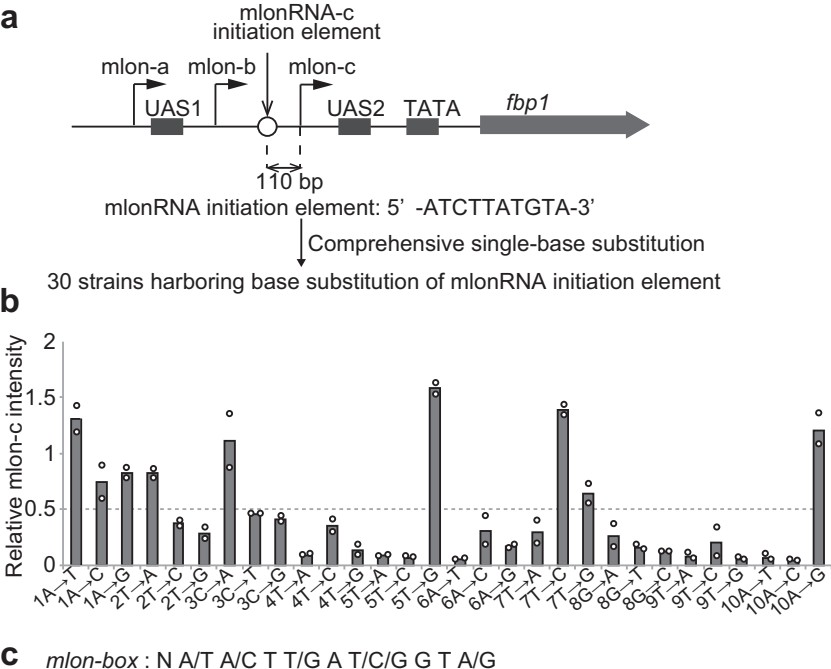

**Fig. 4 Determination of a consensus sequence of *mlon-IE*. a** Schematic representation of the *fbp1* upstream region, including three distinct lncRNAs (mlonRNA-a, mlonRNA-b, and mlonRNA-c) and UAS1 and UAS2. The open circle indicates *mlon-IE* at 110 bp upstream from the mlonRNA-c TSS. The ten nucleotides of the *mlon-IE* (5′-ATCTTATGTA-3′) sequence were comprehensively replaced with each of the three other nucleotides. **b** Relative mlonRNA-c transcript levels of indicated cells at 30 min after glucose starvation compared to wild-type cells. The band intensity of mlonRNA-c transcripts in Supplementary Fig. 6 was quantified. *n* = 2 biologically independent experiments. **c** *mlon-box*, a consensus sequence of *mlon-IE*.

transcription <50% (Fig. 4b and Supplementary Fig. 6), we defined a consensus sequence for *mlon-IE* as 5′-N A/T A/C T T/G A T/C/G G T A/G-3′, and refer to this consensus element as the *mlon-box* (Fig. 4c). Next, we searched for *mlon-box* in the *S. pombe* genome and analyzed the positional relationship between *mlon-box* sites and meiotic recombination sites by using a previously reported Rec12 covalently associated oligonucleotide sequence for identifying meiotic DSB sites[24] (Fig. 5a). This analysis demonstrates that meiotic recombination tends to occur near *mlon-box* sequences. The strongest recombination site that overlapped with the *mlon-box* sequence is located in the intergenic region between *SPBC24C6.09c* and *SPNCRNA.1506* (Fig. 5b). In this region, several meiotic DSBs are observed near the *mlon-box* (Fig. 5c). We inactivated the *mlon-box* by replacing this sequence with a 10 nucleotide sequence from *act1* ORF and to generate *mlon-box-replacement* cells (Fig. 5b)[42]. The *act1* ORF sequence is here used as neutral sequence carrying no DSB activity. In these mutant cells, DSBs closest to the *mlon-box* site are selectively lost (Fig. 5c dotted line, Supplementary Fig. 7a), indicating that the natural *mlon-box* induces meiotic DSB formation. In addition, *SPBC24C6.09c* transcription, but not *SPNCRNA.1506* transcription, is activated early after the onset of meiosis in an *mlon-box*-dependent manner (Fig. 5d and Supplementary Fig. 7b). Also, an MNase-sensitive site appears at the *mlon-box*, and the pattern of MNase-sensitive bands surrounding the *mlon-box* changes in 0.5 and 1 h after the onset of meiosis (Supplementary Fig. 7c). These results indicate that the chromatin configuration around the *mlon-box* changes during meiosis (Supplementary Fig. S7c arrowhead and arrows) and that these changes are dependent on the presence of the *mlon-box* (Supplementary Fig. 7c dotted line). Consistent with these observations, the amount of histone H3 binding at the *mlon-box* decreases in meiosis, and this histone eviction depends on the

presence of the *mlon-box* (Fig. 5e). These results indicate that transcriptional initiation from the natural *mlon-box* in the intergenic region activates meiotic DSB formation via chromatin opening.

Previous genome-wide transcriptome analysis has shown that Atf1 regulates *SPBC24C6.09c* transcription[46]. However, we did not find a canonical Atf1-binding motif (TGACGT) in ±1 kbp from *SPBC24C6.09c-mlon-box*. We noticed a similar palindromic Atf1-binding-like motif (TTACGT: a single G-to-T substitution) in *SPBC24C6.09c-mlon-box* (Supplementary Fig. 8a). In fact, Atf1 binding at *SPBC24C6.09c-mlon-box* increases approximately threefold during meiosis, while the replacement of the entire *mlon-box* (10 bp) results in a pronounced decrease in Atf1 binding (Supplementary Fig. 8b). To selectively disrupt the *mlon-box* while preserving this Atf1-binding sequence, the first three nucleotides (ATC) in the *mlon-box* were replaced with CCT, generating *mlon-box-3bp-replacement* cells (Supplementary Fig. 8a). This strain harboring a mutated *mlon-box* shows little Atf1 binding (Supplementary Fig. 8b). Consistent with this, the transcription level in *mlon-box-3bp-replacement* cells is significantly decreased compared to wild-type cells (Supplementary Fig. 8c). These results indicate the importance of the *mlon-box* in Atf1 binding and later transcriptional activation of *SPBC24C6.09c*. The *SPBC24C6.09c-mlon-box* might enhance Atf1 affinity for the Atf1-binding motif-like sequence. The possible mechanism underlying *mlon-box*-mediated strengthening of Atf1 binding is that mlonRNA molecules initiated from the *mlon-box* bind to the Groucho-Tup1-type transcriptional corepressors Tup11-Tup12 and thereby locally antagonize their repressive functions on Atf1 binding as described previously[47]. An mlonRNA transcription amplification circuit such as this might play a critical role in the massive induction of *SPBC24C6.09c* from an incomplete Atf1-binding site.

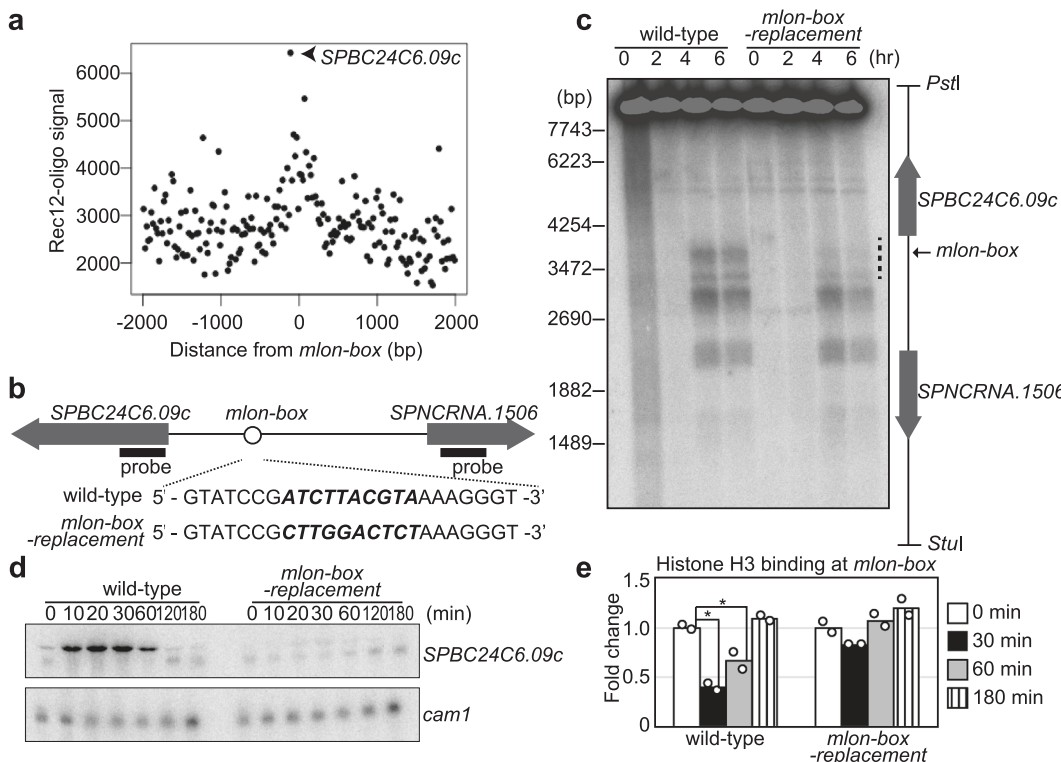

**Fig. 5 DSB formation at the natural meiotic recombination hotspot in the *SPBC24C6.09c* upstream region is mediated by *mlon-box*-dependent intergenic transcription. a** Relationship between *mlon-box* sites and meiotic DSB distribution (Rec12-oligos). The *X* axis indicates the distance from the nearest *mlon-box*, and the *Y* axis indicates copy numbers of the Rec12-oligos-signal. The strongest plot (arrowhead) represents the meiotic recombination hotspot in the *SPBC24C6.09c* upstream region. **b** Schematic representation of the *mlon-box* (open circle) in the intergenic region between *SPBC24C6.09c* and *SPNCRNA.1506*. The entire *mlon-box* sequence comprising 10 nucleotides was replaced with a 10 nucleotide sequence from *act1* ORF in *mlon-box-replacement* cells (bold italic). Location of probes for northern blot was indicated by bold bar. **c** Detection of meiotic DSBs in the intergenic region between *SPBC24C6.09c* and *SPNCRNA.1506*. Indicated cells were cultured and analyzed, as described in Fig. 2b. The dotted line indicates DSB sites around the *mlon-box*. **d** Northern analysis to detect the *SPBC24C6.09c* transcript in wild-type and *mlon-box-replacement* cells. The left probe indicated in Fig. 4b was used. Indicated cells were cultured to induce meiosis, as described in Fig. 2b. The *cam1* transcript is shown as a loading control. **e**, ChIP analysis to examine histone H3 binding at the *mlon-box* in indicated cells. Cells were cultured to induce meiosis, as described in Fig. 2b. The relative increase in the ratio at the indicated time after the onset of meiosis is indicated. *n* = 2 biologically independent experiments. *P*-values were calculated using Student's *t*-test: *$P < 0.05$.

## Discussion

In this study, we identified a novel *cis*-element, the *mlon-box*, which is required for mlonRNA transcriptional initiation. mlonRNA transcription plays a critical role in the massive induction of fbp1 via chromatin opening. In addition, *mlon-box*-mediated transcription activates meiotic DSB formation and recombination in *ade6-M26*, *ade6-4002*, and *ade6-4099* via chromatin opening. Such *mlon-box* transcription-associated meiotic DSB formation is observed in the natural meiotic recombination hotspot located in the *SPBC24C6.09c* upstream region.

It remains to be determined how mlonRNA transcription induces chromatin remodeling. Three mlonRNAs (mlonRNA-a to mlonRNA-c) are initiated far upstream from the *fbp1* promoter and stepwise chromatin remodeling is induced in their transcribed tract[37] (Fig. 1). Each transcriptional initiation efficiently induces local chromatin remodeling only within 290 bp of the initiation site[42], and thus three consecutive mlonRNA transcription events induce stepwise chromatin remodeling over the entire regulatory upstream region of *fbp1*, thereby allowing the binding of transcription factors as well as RNA polymerase II (RNAPII). An intriguing possibility is that the RNAPII that initiates mlonRNA transcription binds unique accessory subunit (s) that possess histone acetyltransferase activity to induce local histone acetylation around these transcription-start sites and dissociate from the initiation complex after promoter clearance[48].

Such histone modifications might recruit an ATP-dependent chromatin remodeler[49–51]. This hypothesis is supported by the observation that histone acetylation is gradually induced from upstream of *fbp1* during glucose starvation and a histone acetyltransferase, Gcn5, and an ATP-dependent chromatin remodeler, Snf22, are required for chromatin remodeling in the upstream from *fbp1*[37,52]. If such novel RNAPII complexes do exist, it will be important to learn how they are selectively targeted to sites of mlon-RNA transcription initiation.

Many lncRNAs are located in the promoter region of genes and referred to as promoter-associated RNAs[53]. *mlon-box* sequences are enriched upstream of TSSs in the *S. pombe* genome (Supplementary Fig. 9), suggesting that *mlon-box*-induced transcription might regulate neighboring gene expression. Consistent with this, several studies have also reported the contribution of ncRNA transcription in gene expression control[7,54,55]. In addition, meiotic recombination sites in fission yeast are directed preferentially to loci expressing ncRNA[13]. It is thus possible that these pervasive lncRNA expressions are involved in the DSB formation as well as with later stages of meiotic recombination via facilitating chromatin remodeling. Taken together, this current study suggests a model that numerous human lncRNAs identified in association with developmental processes[11,12] and diseases including cancer[10,56], are traces of mlonRNA-mediated regulation of genome function. This study provides important insights into the function of widespread lncRNA transcription in the

regulation of DNA-associated reactions on chromatin geometry. A further understanding of the roles of lncRNA transcription might provide important clues to treating lncRNA-associated diseases.

## Methods

**Fission yeast strains, genetic methods, and cell culture.** Supplementary Tables 1 and 2 list the fission yeast strains and primers used in this study, respectively. YEL medium (0.5% yeast extract and 2% glucose), synthetic dextrose (SD) medium, and minimal medium (MM) were used for cell culture[57]. YER medium (yeast extract containing 6% glucose) and YED medium (yeast extract containing 0.1% glucose and 3% glycerol) were used for glucose repression and starvation, respectively. To induce meiosis, diploid cells were grown to a density of $0.5 \times 10^7$ cells/mL in MM medium supplemented with 5 mg/mL of $NH_4Cl$ and 1% YEL medium, then diluted with three times the amount of YEL medium, and further cultured for 4 h. Next, the cells were collected and washed with distilled water twice and transferred to MM medium lacking a nitrogen source. To detect meiotic DSBs, synchronous meiosis using a *pat1-114* mutation was employed together with a *rad50S* mutation to efficiently detect DSBs[58]. The haploid *pat1-114 rad50S* cells were precultured in MM medium containing 5 mg/mL of $NH_4Cl$ at 25 °C, washed with distilled water twice, transferred to MM medium lacking a nitrogen source for arresting the cell cycle at the G1 phase, and cultured further at 25 °C for 16 h. G1-arrested cells were collected by centrifugation, suspended in prewarmed MM medium containing 0.5 mg/mL of $NH_4Cl$ and 1% YEL medium, and cultured at 34 °C to induce meiosis. The transformation was performed using the lithium acetate method as follows[59]. $1 \times 10^8$ cells from exponentially growing culture were collected by centrifugation 3000 rpm. Cells were washed by distilled water and 0.1 M Li-acetate in TE (pH 7.5). Then, cells were resuspended in 40 μl of 0.1 M Li-acetate in TE (pH 7.5), added constructed DNA and 300 μl of 40% PEG 4000 in 0.1 M Li-acetate in TE (pH 7.5). The sample was agitated at 30 °C for 30 min, followed by addition of 40 μl of DMSO and incubation at 42 °C for 15 min. After centrifugation, the cell pellet was resuspended in distilled water and plated on an appropriate selective medium. Mating and sporulation were performed on sporulation agar (SPA) medium[57], followed by random spore analysis.

**Construction of mutant strains carrying mlon-IE 200 bp downstream of ade6-M26, ade6-M375, ade6-4002, ade6-4009, ade6-4095, and ade6-4156.** The *ade6* gene region carrying *M26*, *M375*, *4002*, *4099*, *4095*, or *4156* mutations was amplified using primer sets (p1/p2 and p3/p4). p2 and p4 were flanked by the *fbp1* upstream sequence comprising 65 nucleotides containing *mlon-IE* ($-870$ to $-806$ from the first ATG of *fbp1*) and its complementary sequence, respectively. The resultant products were purified using the QIAquick gel extraction kit (Qiagen, Germany). These fragment pairs were combined at the complementary sequence flanking in p2 and p4 by polymerase chain reaction (PCR) using the p1/p3 primer set and then cloned at *Apa*I and *Spe*I sites in the pBluescript plasmid. Fission yeast cells carrying the *ura4* marker gene in the *Hind*III site in the *ade6* ORF were transformed, and the transformants were selected for uracil auxotrophy using SD plates containing 5-fluoroorotic acid (5-FOA) and uracil.

**Construction of strains with single-base substitution in mlon-IE upstream at fbp1.** Comprehensive *mlon-IE* replacement strains were constructed by PCR amplification according to the method of Senmatsu et al.[42]. Briefly, *fbp1* upstream region was amplified using primers p5/p6 to p35 and p36/p37 to p66. The resultant products were purified using the QIAquick gel extraction kit (Qiagen, Germany). Pairs of fragments were used as templates for PCR amplification using the primer set p5 and p36. Fission yeast cells carrying the *ura4* maker gene in the *Hpa*I site in the *fbp1* promoter were transformed, and the transformants were selected for uracil auxotrophy using SD plates containing 5-fluoroorotic acid (5-FOA) and uracil.

**Construction of strains with mlon-box replacement in SPBC24C6.09c.** For *mlon-box-10bp* replacement in *SPBC24C6.09c*, the *mlon-box* sequence (ATCT-TACGTA) was replaced with the *act1* sequence (CTTGGACTCT) by PCR amplification using primers p67/p68 and p69/p70. For *mlon-box-3bp* replacement in *SPBC24C6.09c*, *mlon-box* sequence was replaced (ATCTTACGTA to CCTTTACGTA) by PCR amplification using primers (p67/p71 and p69/p72).

**Northern blotting.** Northern blotting was performed as follows. Total RNA was prepared from *S. pombe* cells. Briefly, $5 \times 10^7$ cells were suspended in 0.3 ml of RNA extraction buffer (0.5 M NaCl, 0.2 M Tris-HCl [pH 7.5], 0.01 M ethylene-diaminetetraacetic acid (EDTA), 1% SDS) and disrupted with 0.5 g of glass beads and 0.3 ml of phenol-CHCl₃ using multi beads shocker (Yasui kikai, Osaka Japan) at conditions 2300 rpm for 30 s three times. After centrifugation of disrupted materials, *S. pombe* total RNA was isolated from 0.2 ml of supernatant by ethanol precipitation and dissolved in TE buffer (0.01 M Tris-HCl [pH 8.0], 0.001 M EDTA). 10 μg of total RNA was denatured in a buffer (0.02 M MOPS [pH 7.0], 0.005 M sodium acetate, 0.001 M EDTA, 5.7% formaldehyde, 50% formamide) at 60 °C for 5 min. The denatured sample was separated on 1.5% agarose gels containing formaldehyde by electrophoresis in a buffer (0.02 M MOPS [pH 7.0], 0.005

M sodium acetate, 1 mM EDTA) at 100 V for 2 h and transferred to a nylon membrane (Biodyne B, PALL, NY). To detect *ade6* transcripts, an *Eco*RI-*Xho*I restriction fragment of the *ade6* gene was used as a template for random-primer labeling (GE healthcare) with ³²P α-dCTP (PerkinElmer, MA)[31]. To detect *fbp1*, *cam1*, *SPBC24C6.09c*, and *SPNCRNA.1506* transcripts, we amplified the template fragments for random-primer labeling by PCR using primer sets p73/74, p75/76, p77/78, and p79/p80, respectively. Hybridization was performed in a buffer (1% BSA, 7% SDS, 0.5 M $Na_2HPO_4$ [pH 7.4], 1 mM EDTA) at 62 °C for 12 h, and extensively washed with wash buffer (1% SDS, 1 mM EDTA, 0.04 M $Na_2HPO_4$ [pH 7.4]). Signal was detected by a phosphor imager (FLA7000, Fuji film, Tokyo).

**ChIP analysis.** Chromatin immunoprecipitation (ChIP) analyses were performed according to the method of Senmatsu et al.[42] with slight modifications as follows. Fifty ml of culture was incubated with 1.4 ml of 37% formaldehyde solution for 20 min at room temperature, and then 2.5 ml of 2.5 M glycine was added. After centrifugation, collected cells were washed twice with cold TBS buffer (150 mM NaCl, 20 mM Tris HCl [pH 7.5]). The cells were mixed with 400 μl of lysis 140 buffer (0.1% Na-deoxycholate, 1 mM EDTA, 50 mM HEPES-KOH [pH 7.5], 140 mM NaCl, 1% Triton X100) supplemented with protease inhibitor, cOmplete (Roche), and 0.6 ml of zirconia beads were added. After disruption of the cells using a multi beads shocker (Yasuikikai, Osaka), the suspension was sonicated six times for 30 s each on ice to shear chromosomal DNA into around 500 bp fragment, and centrifuged at 4 °C. The supernatant was collected as a whole-cell extract. Two microliters of Anti-Histone H3 antibody (abcam) or 1 μl of Anti-Atf1 antibody (abcam), and 20 μl or 10 μl of Dynabeads Protein A (Invitrogen) were mixed at 4 °C to conjugate antibody and beads, and then washed twice with PBS (138 mM NaCl, 2.7 mM KCl, 10 mM $Na_2HPO_4$, 1.8 mM $KH_2PO_4$) containing 0.1% BSA. Finally, 300 μl of the whole-cell extract was mixed with pretreated beads and allowed to immunoprecipitate at 4 °C for 16 h. The precipitates were washed twice with lysis 140 buffer, once with lysis 500 buffer (0.1% Na-deoxycholate, 1 mM EDTA, 50 mM HEPES-KOH [pH 7.5], 500 mM NaCl, 1% Triton X100), and further washed once with wash buffer (0.5% Na-deoxycholate, 1 mM EDTA, 250 mM LiCl, 0.5% NP-40, 10 mM Tris-HCl [pH 8.0]) followed by once with TE (10 mM Tris-HCl [pH 8.0], 1 mM EDTA). The well-washed precipitates were mixed with 40 μl of elution buffer (10 mM EDTA, 1% SDS, and 50 mM Tris-HCl [pH 8.0]) and allowed to elute the immunoprecipitated protein-DNA complexes at 65 °C for 15 min (immunoprecipitation [IP] sample). To elute IP sample completely, 100 μl of elution buffer and 150 μl of TE containing 0.67% SDS in remaining beads, then incubated at 65 °C for 15 min again. Three microliters of the whole-cell extract was mixed with 97 μl of lysis 140 buffer and 400 μl of TE buffer containing 1% SDS (Input sample). IP and Input sample were added 84 μg of proteinase K (Merck, Darmstadt, Germany), and incubated at 37 °C for 16 hr. After incubation, the temperature was shifted to 65 °C and the sample was further incubated for 6 h. After incubation, DNA was phenol/chloroform extracted from each of the samples and quantified by quantitative PCR using Thermal Cycler Dice Real Time (Takara Bio, Shiga, Japan) and the THUNDERBIRD® SYBR qPCR Mix (TOYOBO, Osaka, Japan). Primer sets p81/p82 at *SPBC24C6.09c-mlon-box* and p83/p84 at the *prp3* locus were used for quantitative PCR analysis.

**Micrococcal nuclease digestion assay.** Analysis of chromatin structure by indirect end-labeling was performed according to the method of Asada et al.[60] with modifications as follows. $5 \times 10^8$ cells from 100 ml of the culture were harvested. Cells were incubated in 0.25 ml of preincubation solution (20 mM Tris-HCl at pH 8.0, 0.7 M 2-mercaptoethanol, 3 mM EDTA) at 30 °C for 10 min and washed once in 1 ml of ice-cold 1 M sorbitol with 10 mM EDTA. Cells were then centrifuged and resuspended in 1 ml of freshly prepared Zymolyase solution (37.5 mM Tris-HCl at pH 7.5, 0.75 M sorbitol, 1.25% glucose, 0.1% (wt/vol) Zymolyase-100T, 6.25 mM EDTA). Cells were incubated for 5 min at 30 °C and resulting spheroplasts were pelleted. All subsequent steps were done at 4 °C. The spheroplasts were then washed once in 1 ml of ice-cold 1 M Sorbitol and resuspended well by pipetting in 1 ml of freshly prepared lysis buffer (18% Ficoll-400, 10 mM $KH_2PO_4$, 10 mM $K_2HPO_4$ [pH 6.8], 1 mM $MgCl_2$, 0.25 mM EGTA, 0.25 mM EDTA, 1 mM PMSF). After centrifugation at 13,000 rpm for 40 min, the crude nuclear pellet was resuspended well in 1.5 ml of buffer A (10 mM Tris-HCl at pH 8.0, 150 mM NaCl, 5 mM KCl, 1 mM EDTA, 1 mM PMSF). 0.5 ml aliquots of the crude chromatin suspension were digested with different amounts of MNase (0, 20, and 50 U/ml) for 5 min at 37 °C in the presence of 5 mM $CaCl_2$. The reaction was terminated by adding 40 mM EDTA, 1% SDS, and 200 μg of proteinase K and incubated at 50 °C for 16 h. Insoluble material was removed by centrifugation at 4 °C. The supernatants were extracted with phenol/chloroform, digested with RNase A, and then extracted once with phenol/chloroform. The extracted DNA was precipitated by 2-propanol, rinsedin 70% ethanol, and resuspended in 50 μl of TE buffer. DNA samples were digested with *Cla*I, *Xho*I or *Apa*I for analysis of *fbp1*, *ade6* or *SPBC24C6.09c* region respectively, and separated by electrophoresis on 1.5% for *fbp1*or 1.2% for *ade6* or *SPBC24C6.09c* agarose gel in Tris base, acetic acid and EDTA (TAE) buffer (0.48% Tris base, 0.11% acetic acid and 0.037% EDTA) at 60 V for 20 h, followed by Southern blotting using the same probe as in the detection of *fbp1*, *ade6* or *SPBC24C6.09c* transcripts. The methods for hybridization and signal detection were the same as in Northern blot analysis.

**Detection of meiotic DSBs**. DNA samples were prepared in agarose plugs from cells from a synchronous meiotic culture according to the method by Hirota et al.[61] with modifications as follows. Briefly, 50 ml of culture was harvested and washed twice using 500 µl of 20 mM citrate-phosphate (pH 5.6), 1.2 M sorbitol, and 40 mM EDTA (CSE) buffer. After resuspension with 500 µl of CSE buffer containing 1.5 mg/mL Zymolyase 20T, cells were incubated at 37 °C for 60 min, centrifuged, and resuspended in 500 µl of 10 mM Tris-Cl (pH 7.5), 0.9 M sorbitol, and 45 mM EDTA (TSE) buffer. An equal volume of low-melting-point agarose (1%) in TSE buffer was added, and the mixture was poured into agarose plug molds (Bio-Rad Laboratories, Hercules, CA, USA). After 5 min on ice, the agarose plugs were collected into a buffer (0.25 M EDTA, 50 mM Tris-Cl [pH 7.5], and 1% sodium dodecyl sulfate [SDS]) and incubated at 55 °C for 90 min. Then, the buffer was removed, and a second buffer (1% lauryl sarcosine, 0.5 M EDTA [pH 9.5], and 1 mg/mL of proteinase K) was added. The mixture was incubated at 55 °C for 2 days. Next, agarose plugs were soaked in 1 ml of Tris-EDTA (TE) containing 0.5 mM phenylmethylsulfonyl fluoride (PMSF) at room temperature for 60 min and washed twice with TE at room temperature for 60 min. The agarose plugs were equilibrated using CutSmart buffer (New England Biolabs, Ipswich, MA, USA) overnight. After changing CutSmart buffer, the agarose plugs were incubated at room temperature for 60 min. DNA in the agarose plugs was digested by AflII or StuI and PstI-HF (New England Biolabs) to detect meiotic DSBs in ade6 or SPBC24C6.09c, respectively. The digested DNA fragments were separated by electrophoresis on 0.8% agarose gel in TAE buffer at 60 V for 20 h, followed by Southern blotting. DNA probes for ade6 and SPBC24C6.09c were amplified by PCR using primer sets p85/p86 and p87/p88, respectively. To detect whole meiotic DSBs, the agarose plugs were washed with TE for 60 min, and the chromosomal DNA and DSBs in agarose plugs were separated by pulse-field gel electrophoresis (PFGE) and stained by ethidium bromide. PFGE was carried out in 0.8% chromosomal grade agarose (Biorad) on a Biorad CHEF-DRIII system and re-circulated at 14 °C. Electrophoresis was performed for 48 h at 2 V/cm in 1 × TAE (Tris-acetate-EDTA) buffer, with a switch time of 30 min at an included angle of 100°.

**Determination of the recombination rate**. The recombination rate was determined according to the method by Hirota et al.[32] with modifications as follows. Briefly, each strain was crossed with a tester strain (ade6-469) on SPA medium at 30 °C for 2 days. Sporulating asci were suspended in 1 ml of distilled water containing 115 units of β-glucuronidase and agitated at 30 °C for 30 min to release spores from asci and kill vegetative cells. After incubation, 500 µl of ethanol was added, and incubation was again performed at room temperature for 15 min. Spores were collected by centrifugation and resuspended in 1 mL of distilled water. Then, $10^4$, $10^3$, and 200 spores were spread onto SD-lacking adenine, SD-lacking leucine and histidine, and YE-containing adenine, respectively. The recombination rate in ade6 was represented as the colony number on SD-lacking adenine normalized by the colony number on YE-containing adenine. The recombination frequency between leu1 and his3 was calculated as the percentage of the colony number on SD-lacking leucine and histidine normalized by the colony number on YE-containing adenine.

**5′ rapid amplification of cDNA ends experiment**. The rapid amplification of complementary DNA (cDNA) ends (RACE) experiment was performed using the SMARTer® RACE 5′/3′ Kit (Takara Bio) according to the manufacturer's instructions. The 5′ ends of the transcripts were amplified by PCR using the universal primer mix included in the kit and the gene-specific primer p89. PCR products were purified using the QIAquick gel extraction kit (Qiagen) and cloned into pCR-BluntII-TOPO (Invitrogen, Carlsbad, CA, USA). Finally, the sequences were determined using the M13 primer p90.

**Statistics and reproducibility**. Throughout the manuscript, all individual data points are plotted. Sample mean and standard deviation are also shown on each bar graph. All statistical analyses were conducted using a one-sided Student's t-test.

**Reporting summary**. Further information on research design is available in the Nature Research Reporting Summary linked to this article.

## Data availability

S. pombe genome sequence and Transcription start sites (TSSs) can be found in PomBase as "Schizosaccharomyces_pombe.ASM294v2.30.dna.genome.fa" and "schizosaccharomyces_pombe.chr.gff3" (2017/3/22, ftp://ftp.pombase.org/pombe).

Rec12 covalently associated oligonucleotide sequence can be found as Rec12-oligo (WT_SOLID_313) under accession No. G7E49977. All uncropped image data used for all figures in the paper are present in Supplementary Fig. 10. Raw data used for all histograms in this paper are present in Supplementary data 1.

## Code availability

Emboss fuzznuc (http://embossgui.sourceforge.net/demo/manual/fuzznuc.html) was used to search mlon-box sequences.

Bedtools (v.2.17.0; https://github.com/arq5x/bedtools2) was used for enrichment analysis around TSS.

R code (https://www.r-project.org/) used for visualization in this study is available at https://www.datamentor.io/r-programming/histogram/.

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

## Acknowledgements

We thank Dr. Walter W. Steiner (Department of Biology, Niagara University) for the gift of ade6-4002, −4099, −4095, and −4156 background cells. We also thank all the members of the Hirota laboratory for their help. We acknowledge the Radioisotope Research Center in Tokyo Metropolitan University for support in the use of isotopes. This work was supported in part by JSPS KAKENHI (16H01314 to K.H. and 19J20773 to S.S.) and the Takeda Science Foundation and Yamada Science Foundation (to K.H.). Correspondence and requests for materials should be addressed to Kouji Hirota (khirota@tmu.ac.jp).

## Author contributions

Conceived and designed the experiments: S.S., R.A. and K.H. performed most of the experiments: S.S. Performed the bioinformatics analysis: O.A. Wrote the paper: S.S., R.A., C.S.H., K.O. and K.H.

## Competing interests

The authors declare no competing interests.
