## [Peer Review File · Communications Biology]

Reviewers' comments:

Reviewer #1 (Remarks to the Author):

Comments for "lncRNA transcription induces meiotic recombination via chromatin remodeling"

In this study, the author identified a novel cis-element, the mlon-box, which is required for mlonRNA transcription initiation. mlonRNA transcription plays an important role in meiotic DSB formation and recombination at meiotic hot spot ade6-M26, and SPBC24C6.09c upstream intergenic region. The authors suggest that the intergenic lncRNA transcription is responsible for meiotic DSB induction via chromatin remodeling. The identification of mlon-box element is an impressive finding, and the conclusions presented are interesting. This reviewer suggests addressing following points:

1. Fig. 1c, in light of author's previous work, I wonder whether glucose starvation can induce mlonRNA expression at M26 site? Moreover, it would be interesting to know whether mlonRNA expression can further induce DSB under this condition.
2. Fig. 2a, the authors shall include M26 without mlon-IE as a control. This is important because the level of M26 transcription is considerably lower in M26::mlon-IE-mut as compared to WT M26 (Fig.1c).
3. The experiments to study the effect of mlon-IE have been performed in the context of strong meiotic DSB hot spot (ade6 locus or SPBC24C6.09c). Perhaps they shall also investigate whether mlon-IE could transfer a meiotic DSB cold spot to a hot spot?
4. It would be nice to know whether ade6-4095::mlon-IE or ade6-4156::mlon-IE show meiotic DSB formation?
5. Fig. 3b, the expression of mlon-IE-mut should be included.
6. Fig. 5b, the orientation of mlon-box is different at gene SPBC24C6.09c as compared to other examples included. Does the direction of mlon RNA makes any difference in inducing chromatin changes and DSB formation?

Reviewer #2 (Remarks to the Author):

This manuscript by Senmatsu et al. explores the connection between long non-coding (lnc)RNA expression and the control of meiotic recombination. The authors reveal that, by using classical assays to monitor DNA break formation and recombination during fission yeast meiosis, that lncRNA transcription stimulates meiotic recombination, ostensibly via an induction of DNA break formation. There is already a firm connection between transcriptional activity and DNA break activity in meiosis. As such, these data reaffirm this notion, but also expand our knowledge of this process and invoke a novel role for non-coding RNA transcription in this process. In general, this work is well-done (with some caveats, as outlined below), and makes several interesting observations. I support eventual publication of this work, after several issues have been addressed. In terms of experimental points raised, point 7, 8 and 11 are the most crucial.

- 1) In general, I feel that the authors need to give more attention to previous literature on the connection between transcription and meiotic recombination. For example, the introduction should include a paragraph outlining why the authors decide to focus on this process: what is the impetus for doing this, what do we know about the regulation of meiotic recombination (by active transcription) in fission yeast and other systems? The same holds for the discussion; please put the work in context better, especially in relation to what we know about the lncRNAs in meiotic regulation, and the

connection between transcription and DSB activity. For the uninitiated reader, it remains difficult to digest the connections here.

2) In addition, I would advise the authors to also include/discuss the possibility that recombination rates are changed independently of DSB activity (i.e. lncRNA-induced chromatin remodeling could have effect post-DSB formation). Although the correlation between DSB activity and recombination is tempting, other things can happen downstream in the repair process, which can influence crossover frequencies.

3) Line 19: "the cis-element for lnc-RNA transcription" was already shown in a previous paper (Senmatsu et al. 2019, Scientific Reports); please rephrase.

4) Line 168: why was act1 ORF sequence chosen? Any specific reason that it is DSB neutral? Could be mentioned in one line.

5) Fig 1a: can an mNase-treated chromatin remodelling blot be included beside the glucose starvation time course to show how the gradual increase in mlonRNA during meiotic induction results in chromatin remodelling?

6) Is the endogenous FBP1 promoter region deleted in these cells or are we also looking at the mlonRNA induced from that site? Are Atf1 and Rst2 constitutively active, which is why we see a M26 band at M26 (0hr)? Please elaborate and explain better.

7) To be able to claim effects by mlon-IE/melon-IE-mut on M26 recombination levels, it is essential to include recombination in the parental M26 background (Figure 2a and b). Same for M375. Without these data the current claim cannot be substantiated. This is an important point that needs to be addressed.

8) For all southern blots (for example fig 2b, 4d, e): To be able to more strongly claim that the effect is local, can the authors provide evidence that DSB activity elsewhere in the genome is not altered by these manipulations? Southern blot analyses to show activity at another (not modulated) DSB site would be ideal.

9) Why would the MNase treated band intensity change at M26 in M26::mlon-IE compared to the M26 condition? Shouldn't this remain unchanged?

10) Fig 5A: can the authors provide a control plot comparing all the rec12-oligos to provide insight into what percentage of rec12 oligos coincide with or are present near mlon box sites from all possible rec12 DSB sites?

11) Fig 5C: please repeat, and quantify DSB activity (of total DNA signal). Total signal on this Southern Blot seems lower for the mlon-box-replacement situation (especially when comparing parental DNA signals), and this makes making any conclusions on DSB intensity difficult. Quantification (of DSBs from multiple SB analyses) is important to demonstrate that the DSB signals are really lower, as claimed by the authors.

Reviewer: Gerben Vader

Responses to Reviewers

Reviewer #1 (Remarks to the Author):

Comments for “lncRNA transcription induces meiotic recombination via chromatin remodeling”

In this study, the author identified a novel cis-element, the mlon-box, which is required for mlonRNA transcription initiation. mlonRNA transcription plays an important role in meiotic DSB formation and recombination at meiotic hot spot ade6-M26, and SPBC24C6.09c upstream intergenic region. The authors suggest that the intergenic lncRNA transcription is responsible for meiotic DSB induction via chromatin remodeling. The identification of mlon-box element is an impressive finding, and the conclusions presented are interesting. This reviewer suggests addressing following points:

1. Fig. 1c, in light of author’s previous work, I wonder whether glucose starvation can induce mlonRNA expression at M26 site? Moreover, it would be interesting to know whether mlonRNA expression can further induce DSB under this condition.

(Response) The reviewer raises an interesting point. We detected mlonRNA expression at M26 site in response to osmotic stress, but not to glucose starvation stress. Glucose starvation induces activation of both Atf1 and Rst2, and thus only genes (such as fbp1) carrying binding sites for both transcription factors can respond.

Spo11 (Rec12 in *S. pombe*) digests DNA in meiosis and is responsible for meiotic DSB formation, and the expression of *REC12* is strictly regulated not to express in mitotic cell cycle. Thus, glucose starvation in mitotic cell-cycle cannot induce DSB at M26.

2. Fig. 2a, the authors shall include M26 without mlon-IE as a control. This is important because the level of M26 transcription is considerably lower in M26::mlon-IE-mut as compared to WT M26 (Fig.1c).

(Response) Thank you for this advice. To comply with this comment, we added the data for M26 without mlon-IE as a control. Insertion of mlon-IE drastically changes amino acid sequence of Ade6, thereby disrupting the function of Ade6 protein (shown in new supplementary figure S2). Thus, we cannot directly compare the number of Ade+ recombinants between M26 cells and mlon-IE or mlon-IE-mut inserted cells. This is

because the distance between *ade6-M26* and *ade6-469* (tester cells) is longer than the distance between the inserted *mlon-IE* and *ade6-469*. Moreover, *ade6-M26::mlon-IE* has multiple mutations and longer recombination tracts in meiotic recombination are required for the reconstitution of functional *ade6+* gene in the cross between *ade6-M26::mlon-IE* and *ade6-469* than that in *ade6-M26* and *ade6-469*.

3. The experiments to study the effect of *mlon-IE* have been performed in the context of strong meiotic DSB hot spot (*ade6* locus or SPBC24C6.09c). Perhaps they shall also investigate whether *mlon-IE* could transfer a meiotic DSB cold spot to a hot spot?

(Response) We put *mlon-IE* at 200 bp downstream of *ade6-M375* (coldspot control) and demonstrated that *mlon-IE* at *ade6-M375* has no effect (fig. 2c). This result suggests that *mlon-IE* cannot work alone and requires the assistance of other transcription factors to induce *lnc-RNA* expression and chromatin remodeling.

4. It would be nice to know whether *ade6-4095::mlon-IE* or *ade6-4156::mlon-IE* show meiotic DSB formation?

(Response) Thank you for this kind comment. We analyzed the effects of *mlon-IE* in *ade6-M26*, *ade6-4002* (CCAAT-motif) and *ade6-4099* (oligo-C-motif), *ade6-4095*, and *ade6-4156*, and found that *mlon-IE* works with these putative TF binding motifs and induces *lnc-RNA* transcription (fig. 3 a,b). We then decided to further analyze the effect of *mlon-IE* in meiotic DSB formation in *ade6-M26*, *ade6-4002* (CCAAT-motif) and *ade6-4099* (oligo-C-motif), but not in *ade6-4095* or *ade6-4156*, because the associated transcription factors for these elements (*ade6-4095* or *ade6-4156*) had not been identified and it seemed less informative to know *mlon-IE* plays roles in meiotic DSB formation in the collaboration with unknown factors. Please understand this situation.

5. Fig. 3b, the expression of *mlon-IE-mut* should be included.

(Response) Thank you for this comment. We showed that the additional *lncRNA* was expressed at the downstream of *ade6-M26* from inserted *mlon-IE*. We found that this *lncRNA* expression is induced by the specific sequence of *mlon-IE* but not by fragment insertion itself as evidenced by the observations that *mlon-IE-mut* has no effect in the same insertion condition. Similarly, we demonstrated that mutations of the natural *mlon-BOX* in SPBC24C6.09c also extinguished expression from this site as well as the

subsequent meiotic DSB formation at this site. These data indicate that the specific sequence of mlon-IE, and not just any fragment insertion itself, affects lncRNA expression.

6. Fig. 5b, the orientation of mlon-box is different at gene SPBC24C6.09c as compared to other examples included. Does the direction of mlon RNA makes any difference in inducing chromatin changes and DSB formation?

(Response) Like the other cis-element including CRE sequence, the mlon-BOX may also play roles in both orientations as evidenced by this observation. In the case of lncRNA expression at SPBC24C6.09c upstream, the direction of transcription is the same to the gene SPBC24C6.09c, but not to the gene SPNCRNA.1506 as evidenced by the data shown in fig.5d and fig.S7.

Reviewer #2 (Remarks to the Author):

This manuscript by Senmatsu et al. explores the connection between long non-coding (lnc)RNA expression and the control of meiotic recombination. The authors reveal that, by using classical assays to monitor DNA break formation and recombination during fission yeast meiosis, that lncRNA transcription stimulates meiotic recombination, ostensibly via an induction of DNA break formation. There is already a firm connection between transcriptional activity and DNA break activity in meiosis. As such, these data reaffirm this notion, but also expand our knowledge of this process and invoke a novel role for non-coding RNA transcription in this process. In general, this work is well-done (with some caveats, as outlined below), and makes several interesting observations. I support eventual publication of this work, after several issues have been addressed. In terms of experimental points raised, point 7, 8 and 11 are the most crucial.

1) In general, I feel that the authors need to give more attention to previous literature on the connection between transcription and meiotic recombination. For example, the introduction should include a paragraph outlining why the authors decide to focus on this process: what is the impetus for doing this, what do we know about the regulation of meiotic recombination (by active transcription) in fission yeast and other systems? The

same holds for the discussion; please put the work in context better, especially in relation to what we know about the lncRNAs in meiotic regulation, and the connection between transcription and DSB activity. For the uninitiated reader, it remains difficult to digest the connections here.

(Response) Thank you for this comment. We added detailed descriptions in the introduction about the relationship between lncRNAs and meiotic recombination and meiotic recombination (DSB) regulations associated with chromatin as well as RNA expression.

2) In addition, I would advise the authors to also include/discuss the possibility that recombination rates are changed independently of DSB activity (i.e. lncRNA-induced chromatin remodeling could have effect post-DSB formation). Although the correlation between DSB activity and recombination is tempting, other things can happen downstream in the repair process, which can influence crossover frequencies.

(Response) Thank you for this advice. We discussed the points you kindly raised.

3) Line 19: “the cis-element for lnc-RNA transcription” was already shown in a previous paper (Senmatsu et al. 2019, Scientific Reports); please rephrase.

(Response) Thank you for this advice. We rephrased this text. (in abstract)

4) Line 168: why was act1 ORF sequence chosen? Any specific reason that it is DSB neutral? Could be mentioned in one line.

(Response) Thank you for this comment. We explained the reason why we used act1 sequence in the text. (page 9, line 26)

5) Fig 1a: can an mNase-treated chromatin remodelling blot be included beside the glucose starvation time course to show how the gradual increase in mlonRNA during meiotic induction results in chromatin remodelling?

(Response) Thank you for this advice. We added the data showing the changes of MNase digestion pattern during glucose starvation in the new fig.1b.

6) Is the endogenous FBP1 promoter region deleted in these cells or are we also looking at the mlonRNA induced from that site? Are Atf1 and Rst2 constitutively active, which is why we see a M26 band at M26 (0hr)? Please elaborate and explain better.

(Response) mlonRNAs in *fbp1* upstream region and the lncRNA expressed from mlon-IE inserted *ade6-M26* are distinctly different molecules. lncRNA expression from mlon-IE, but not the molecules themselves have an impact on the chromatin around initiation site. We accordingly explain in the text.

7) To be able to claim effects by mlon-IE/melon-IE-mut on M26 recombination levels, it is essential to include recombination in the parental M26 background (Figure 2a and b). Same for M375. Without these data the current claim cannot be substantiated. This is an important point that needs to be addressed.

(Response) Thank you for this advice. To comply with this comment, we added the data for M26 without mlon-IE as a control. Insertion of mlon-IE drastically changes the amino acid sequence of the Ade6 protein, disrupting Ade6 activity (shown in new supplementary figure S2). Thus, we cannot directly compare the number of *ade+* recombinants between M26 cells and mlon-IE or mlon-IE-mut inserted cells. This is because, the distance between *ade6-M26* and *ade6-469* (tester cells) is longer than the distance between inserted mlon-IE and *ade6-469*. Moreover, *ade6-M26::mlon-IE* has multiple mutations and longer recombination tracts in meiotic recombination are required for the reconstitution of functional *ade6+* gene in the cross between *ade6-M26::mlon-IE* and *ade6-469* than that in *ade6-M26* and *ade6-469*.

8) For all southern blots (for example fig 2b, 4d, e): To be able to more strongly claim that the effect is local, can the authors provide evidence that DSB activity elsewhere in the genome is not altered by these manipulations? Southern blot analyses to show activity at another (not modulated) DSB site would be ideal.

(Response) Thank you for this advice. We analyzed whole DSBs using pulse field gel electrophoresis (PFGE), and confirmed that DSB activity in the whole genome is not altered by these manipulations. (new figure S4)

9) Why would the MNase treated band intensity change at M26 in M26::mlon-IE compared to the M26 condition? Shouldn't this remain unchanged?

(Response) It is possible that the chromatin at the M26 site becomes less open in response to the shift of the chromatin status into an open configuration at the mlon-IE inserted site. We discuss the possibility that local geometry of chromatin configuration could influence the adjacent chromatin status. (page8, line1)

10) Fig 5A: can the authors provide a control plot comparing all the rec12-oligos to provide insight into what percentage of rec12 oligos coincide with or are present near mlon box sites from all possible rec12 DSB sites?

(Response) We already showed all cumulative Rec12-oligo numbers falling in -2000 to +2000 bp from mlon-BOX sequence.

11) Fig 5C: please repeat, and quantify DSB activity (of total DNA signal). Total signal on this Southern Blot seems lower for the mlon-box-replacement situation (especially when comparing parental DNA signals), and this makes making any conclusions on DSB intensity difficult. Quantification (of DSBs from multiple SB analyses) is important to demonstrate that the DSB signals are really lower, as claimed by the authors.

(Response) We have repeated the experiment and quantified DSB intensity (new fig. 5c and fig.S7).

REVIEWERS' COMMENTS:

Reviewer #1 (Remarks to the Author):

All the points raised by this reviewer have been adequately addressed. I support publication of the manuscript

Reviewer #2 (Remarks to the Author):

In general the reviewers have done a careful and commendable job in the revision of this manuscript. There is one minor point that remains to be addressed before this manuscript can be accepted. As requested by this reviewer, the other have compared Mlon-ie, Mlon-ie-mut to M26 (Figure 2), and the authors point out that the recombination rates cannot be compared due too several reasons, which are all reasonable. However, just the fact that mlon-ie-mut shows lower recombination rates as compared to Mlon-ie cannot by itself not be taken as proof that Mlon-ie "stimulates meiotic recombination" (page 7, line 53). I therefore urge the authors to scale down any claims here regarding stimulation of recombination. Later experiments (for example Figure 5) provide good proof that the MLon sequence is required for recombination, so the main conclusion can stand, but conclusions regarding the recombination analysis as shown in Figure 2 need to be adjusted. Please rephrase the text.

Responses to Reviewers

Reviewer #1 (Remarks to the Author):

All the points raised by this reviewer have been adequately addressed. I support publication of the manuscript

(Response) Thank you very much for your kind comments and supports. With your critical comments and the resulting changes, our manuscript has been significantly improved

Reviewer #2 (Remarks to the Author):

In general the reviewers have done a careful and commendable job in the revision of this manuscript. There is one minor point that remains to be addressed before this manuscript can be accepted. As requested by this reviewer, the other have compared Mlon-ie, Mlon-ie-mut to M26 (Figure 2), and the authors point out that the recombination rates cannot be compared due too several reasons, which are all reasonable. However, just the fact that mlon-ie-mut shows lower recombination rates as compared to Mlon-ie cannot by itself not be taken as proof that Mlon-ie "stimulates meiotic recombination" (page 7, line 53). I therefore urge the authors to scale down any claims here regarding stimulation of recombination. Later experiments (for example Figure 5) provide good proof that the Mlon sequence is required for recombination, so the main conclusion can stand, but conclusions regarding the recombination analysis as shown in Figure 2 need to be adjusted. Please rephrase the text.

(Response) Thank you very much for your kind comment here. According to this kind advice, we rephrased this text as follows.

‘These findings indicate that *m lon-IE* induces mlonRNA transcription even at 200 bp downstream of the *M26* mutation point in the *ade6-M26* gene and that this expression may augment meiotic recombination in comparison to mutated *m lon-IE* inserted cells.’